# Reduced Neuroinflammation and Improved Functional Recovery after Traumatic Brain Injury by Prophylactic Diet Supplementation in Mice

**DOI:** 10.3390/nu11020299

**Published:** 2019-01-31

**Authors:** Jin Yu, Hong Zhu, Saeid Taheri, William L. Monday, Stephen Perry, Mark S. Kindy

**Affiliations:** 1Department of Pharmaceutical Sciences, College of Pharmacy, University of South Florida, Tampa, FL 33620, USA; jinyu@health.usf.edu (J.Y.), hongzhu@health.usf.edu (H.Z.), taheris@health.usf.edu (S.T.), wmondy@health.usf.edu (W.L.M.); 2NutriFusion®, LLC, Naples, FL 34109, USA; sperry@consealint.com; 3Departments of Molecular Medicine, Molecular Pharmacology, Physiology and Pathology and Cell Biology, and Neurology, College of Medicine, University of South Florida, Tampa, FL 33620, USA; 4James A. Haley VA Medical Center, Tampa, FL 33612, USA; 5Shriners Hospital for Children, Tampa, FL 33612, USA

**Keywords:** diet, traumatic brain injury, inflammation, mitochondrial biogenesis, behavior

## Abstract

Currently, there are no approved therapeutic drugs for the treatment of traumatic brain injury (TBI), and new targets and approaches are needed to provide relief from the long-term effects of TBI. Recent studies suggest that nutrition plays a critical role in improving the outcome from TBI in both civilians and military personnel. We have previously shown that GrandFusion^®^ (GF) diets improved recovery from cerebral ischemia and enhanced physical activity and endurance in rodent models. We, therefore, sought to determine the impact of a prophylactic diet enriched in fruits and vegetables on recovery from TBI in the controlled cortical impact rodent model. Results demonstrated that mice fed the diets had improved neuromotor function, reduced lesion volume, increased neuronal density in the hippocampus and reduced inflammation. As previously shown, TBI increases cathepsin B as part of the inflammasome complex resulting in elevated inflammatory markers like interleukin-1β (IL-1β). Consumption of the GF diets attenuated the increase in cathepsin B levels and prevented the increase in the proapoptotic factor Bax following TBI. These data suggest that prior consumption of diets enriched in fruits and vegetables either naturally or through powdered form can provide protection from the detrimental effects of TBI.

## 1. Introduction

Traumatic brain injury (TBI) is the consequence of an external impact that triggers pathways contributing to pathological changes in the brain leading to alterations in brain function [1]. TBI is a serious health issue with more than 1.8 million Americans affected each year [2]. Common causes of TBI or concussions are not limited to combat-related injuries, but most often occur from incidents associated with motor vehicle collisions, falls, sports, and assaults [3]. A TBI occurs when a sudden blow causes the brain to hit the skull. The result can be a mild, moderate or severe brain injury and the signs and symptoms can be hard to recognize [4,5]. Sustaining a concussion or TBI can lead to changes in cognitive abilities and control of emotions, mobility, speech, and senses [6]. Undiagnosed and untreated, a TBI can have a huge impact on how a person thinks and acts, and on his or her mental health.

To date, there are no good treatments for TBI. While a therapy for this deleterious disorder still remains elusive, there has been a tremendous effort in translational research to understand and manage the clinical symptoms subsequent to TBI [7,8]. A number of studies have implicated dietary supplements for both prevention and treatment of brain syndromes [9,10,11]. The use of omega-3 polyunsaturated fatty acids (*ω*-3 PUFAs) has been explored extensively for ischemic injury, Alzheimer’s, and Parkinson’s disease, as well as TBI [12]. Dietary supplementation with *ω*-3 PUFAs in humans has been shown to be remarkably safe and can be consumed over long periods prophylactically. In addition, nutrition appears to be a significant predictor of death due to TBI [13]. Together with the prevention of arterial hypotension, hypoxia, and intracranial hypertension, it is one of the few therapeutic interventions that can directly affect TBI outcomes. After a TBI, early initiation of nutrition is recommended. 

GrandFusion^®^ (GF) are blends of fruits and vegetables, vastly enhanced with vitamins and nutrients that are able to attenuate the degree of cerebral ischemia injury and limit several parameters of stroke, such as inflammatory markers and reactive oxygen species and behavioral changes [14]. Furthermore, GF blends can improve memory and learning in aged rats and improve physical activity mediated by antioxidant enzymes and signaling pathways [15,16]. Prior research has shown that GF has anti-inflammatory, anti-oxidant, neuroprotective and neurogenic properties [14,15].

In the current study, our goal was to determine the influence of prophylactic diets rich in vegetables and fruits on the outcomes associated with traumatic brain injury (TBI). Mice were fed diets enriched in fruits and vegetables for 2 months prior to and then subjected to TBI. The objective of the study was to determine if the presence of these nutraceutical and phytochemicals can limit the extent of the injury following TBI. The results divulged that these diets were able to attenuate the damage elicited by the TBI. Behavioral changes, inflammation, lesion volume, and proapoptotic markers were examined in mice chronically exposed to the diets. Finally, we have demonstrated that supplementation of mice with the enhanced diets limited the extent of the injury, reduced inflammation and altered pathways critical to the injury process. These data suggest that these prophylactic diets can influence the pronounced changes seen in TBI to decelerate the process and improve outcome.

## 2. Materials and Methods

### 2.1. Animal Experiments

C57BL/6 mice (Jackson Laboratory, Bar Harbor, ME, USA), weighing 22 to 25 g each were given free access to food and water before the experiment. Animals (100 male mice) were 10 to 12 weeks of age at the start of the experiment and were maintained on a 12-hour light/dark cycle (lights on at 7:00 a.m.). All animals were randomized to the various groups. Prior to TBI, animals were fed for 2 months a normal diet or a normal diet with ~2% supplementation of the different materials NF-216 (GrandFusion–Fruit and Veggie #1 Blend), NF-316 (GrandFusion–Fruit #2 Blend), and NF-416 (GrandFusion–Vegetable #3 Blend) [14,15,16]. See Appendix A for the composition of supplementation. Animals were gavaged with the supplements on a daily basis, once per day. GrandFusion supplements were prepared by NutriFusion, LLC (www.nutrifusion.com). Average food intake was 3.75 ± 0.07 g/day/mouse, and the average consumption of diets was 0.08 ± 0.005 g/day/mouse. All studies were approved that the Institutional Animal Care and Use Committee at the Medical University of South Carolina and the Veterans Affairs Medical Center. This study adhered to the Guide for the Care and Use of Laboratory Animals developed by the Office of Laboratory Animal Welfare.

### 2.2. TBI Injury

The controlled cortical impact (CCI) mouse model was used to deliver a controlled, consistent injury to all animals [17]. The procedure requires surgical removal and replacement of a portion of the skullcap to be able to directly injure the brain. Adult mice were anesthetized with ketamine (90 mg/kg) and xylazine (10 mg/kg) administered by intraperitoneal injection of 0.02 mL of solution per gram of body weight. The degree of anesthesia was assessed by testing corneal reflexes and toe-pinch reflexes. During anesthesia, mice were placed in a stereotaxic frame, with the head positioned in the horizontal plane and the nose bar set at −5. Using sterile procedures (site was shaved and cleaned with Wescodyne before surgical manipulation), the head was positioned in the horizontal plane with the nose bar set at zero. After a mid-line incision exposing the skull, a 3-mm craniotomy was made on the right side of the brain lateral to the sagittal suture and centered between lambda and bregma. The skull at the craniotomy site was removed without disrupting the underlying dura. The exposed cortex was injured using a CCI device (Precision Systems and Instrumentation, Fairfax Station, VA, USA) armed with a 2-mm tip. The CCI device was set at a velocity of 3.5 m/sec and to a depth of 2 mm, with a dwell time of 100 ms. After injury, a small, round cover glass was placed on the skull to cover the injury site (EMS no. 72296-05, 5 mm diameter, #1.5 thickness; Electron Microscopy Sciences, Hatfield, PA, USA). The glass was not glued down, and within a few hours, the glass was covered by matrix that prevented movement. The cover glass was autoclaved before use. This helped prevent tearing and sticking of the scalp to the injury site. The cover glass did not affect swelling. The skin was then stapled together, and the animals were placed on a heating pad to recover. Total surgical time was less than 45 min. The survival rate for this procedure was approximately 90%. Animals were returned to their home cages after recovery from anesthesia and monitored daily for any signs of discomfort or other abnormal behavior, and none were observed.

### 2.3. Cathepsin B Activity Assay

Brain cathepsin B activity was measured 2 h after trauma using a fluorometric assay kit, as described by the manufacturer (ab65300; Abcam, Cambridge, MA, USA). Briefly, tissues were washed twice in ice-cold phosphate-buffered saline and then homogenized in extraction buffer, as described by the manufacturer. After 10-min incubation on ice, the extract was centrifuged at 10,000 *g* for 5 min, and 50 μL of supernatant was mixed with an equal volume of 2 × reaction buffer and 2 μL of substrate in a 96-well microplate. Plates were kept in the dark at 37 °C for 1 h, and fluorescence was recorded using a FLUOstar Optima plate reader (BMG LABTECH GmbH, Ortenberg, Germany). Protein concentration was determined by the bicinchoninic acid assay method (Bio-Rad, Hercules, CA, USA). Cathepsin B activity was measured in triplicate and was expressed as fluorescent units/mg of protein. For the determination of enzyme activity, we isolated the region of trauma for analysis.

### 2.4. Cathepsin B and Bax Western Blot Analyses

Brain cathepsin B, Bax, and actin (control) protein levels were determined 24 h after sham operation or TBI, because cathepsin B and Bax protein levels are known to be significantly increased at that time post-TBI [17]. Relative levels of cathepsin B, Bax, and actin in the supernatant fraction from the brain extract were determined by Western blot (polyclonal antibodies: Cathepsin B, sc-13985; Bax, sc-526; β-actin, sc-130657; Santa Cruz Biotechnology, Santa Cruz, CA, USA), as described previously [18]. Relative intensities of Western blot bands were assessed by densitometry in triplicate for each sample. Densitometric analysis was done using IQTL (Imagequant TL) software (GE Life Sciences, Piscataway, NJ, USA). For protein studies, the entire lesioned area was harvested for Western blot analysis. In control or sham animals, a similar region was harvested.

### 2.5. ELISA Analysis

For quantitative analysis of cytokines, an ELISA was used to measure the levels of tumor necrosis factor-α (TNF-α), interleukin-1β (IL-1β), or transforming growth factor-β (TGF-β) in brain tissue [19]. Cytokines were extracted from mouse brains as follows: frozen hemibrains were placed in tissue homogenization buffer containing protease inhibitor cocktail (Sigma, St Louis, MO, USA) 1:1000 dilution immediately before use, and homogenized using polytron. Tissue sample suspensions were distributed in aliquots and snap frozen in liquid nitrogen for later measurements. Invitrogen ELISA kits were then used, according to manufacturer directions (Carlsbad, CA, USA).

### 2.6. Rotarod Assay

An automated rotarod (San Diego Instruments, San Diego, CA, USA) was used to assess the effects on vestibulomotor function of mice after trauma [20]. On the day preceding injury, mice underwent two consecutive conditioning trials at a set rotational speed (16 revolutions per min) for 60 sec, followed by three additional trials with accelerating rotational speeds. The average time to fall from the rotating cylinder in the latter three trials was recorded as baseline latency. After injury, mice underwent consecutive daily testing with three trials of accelerating rotational speed (inter-trial interval of 15 min). Average latency to fall from the rod was recorded. Mice unable to grasp the rotating rod were given a latency of 0 sec. The experimenter was blinded as to the groups of animals.

### 2.7. Wire Hanging Test

The wire hanging apparatus was comprised of a stainless-steel bar (50 cm; 2 mm diameter), resting on two vertical supports and elevated 37 cm above a flat surface. This test was performed as previously described by researchers blinded to the experimental groups [21].

### 2.8. Grid Walking and Foot-Fault Test

The grid walking test is sensitive to deficits in descending motor control [22]. Each mouse was placed on a stainless-steel grid floor (20 × 40 cm with a mesh size of 4 cm^2^) elevated 1 m above the floor. For a videotaped 1-minute-long observation period, the total number of steps was counted. The number of foot-fault errors (when the animals misplaced a forelimb or hind limb such that it fell through the grid) was also recorded for 1 minute.

### 2.9. Cylinder Test and the Morris Water Maze Test

The cylinder test and the Morris Water Maze tests were carried out as previously described by researchers blinded to the experimental groups [23,24]. In the cylinder test, a total of 20 movements were recorded during the 10-minute test. The final score was determined based on the following formula: final score = (non-impaired forelimb movement − impaired forelimb movement)/(non-impaired forelimb movement + impaired forelimb movement + both movements)(1)

This test evaluates forelimb use asymmetry for weight shifting during vertical exploration and provides high reliability even with inexperienced raters. Occasionally, mice with large deficits did not move frequently enough to obtain an adequate number of vertical movements. Typically, these mice would recover in time when the test was performed. To avoid bias, these mice were not scored until they could perform the test. These tests were carried out by researchers blinded to the study groups.

### 2.10. Brain Lesion Volume Analysis

Histological analysis occurred on the last day of the behavioral assay (day 35 post-TBI mice) to allow correlation of behavior with pathology [17]. Mice were anesthetized and transcardially perfused with saline and 10% buffered formalin phosphate solution containing 4% paraformaldehyde (PFA). Brains were removed, postfixed in PFA for 24 h, and protected in 30% sucrose. Frozen brain sections (30 μm) were cut on a cryostat and mounted onto glass slides. Every fourth section was processed for immunohistochemical analysis beginning from a random start point before the lesioned area. Thirty-micron sections were stained with hematoxylin and eosin (H&E), dehydrated, and mounted for analysis. Lesion volume in each section was determined with a computer-assisted image analysis system, consisting of a Power Macintosh computer (Apple Inc., Cupertino, CA, USA) equipped with a QuickCapture frame grabber card, Hitachi CCD camera (Hitachi Kokusai Electric Inc., Tokyo, Japan) mounted on an Olympus microscope (Olympus, Tokyo, Japan), and camera stand. Images were captured, and the total area of damage was determined over sections using National Institutes of Health (NIH) Image Analysis Software (v. 1.55; NIH, Bethesda, MD, USA) conducted by a single operator blinded to treatment status for analyses of all measurements.

### 2.11. Neuronal Cell Density Determination

Cell counting was conducted using a Nikon Eclipse E800 light microscope (Nikon Imaging Japan Inc., Tokyo, Japan) interfaced with the StereoInvestigator software package (MicroBrightField, Williston, VT, USA) [17]. Neuronal density was calculated as the number of stained neurons per volume of hippocampus determined using the optical fractionator method, as previously described [25,26,27]. Before counting, all slides were coded to avoid bias. As determined by StereoInvestigator, three sections (40 μm) spaced eight sections apart along the hippocampal formation were selected by systematic random sampling. On each section, the hippocampal area was delineated. Only cells within the counting frame or overlapping the right or superior border of the counting frame, and for which nuclei came into focus while focusing down through the dissector height, were counted. Tissue generated and H&E labeled for the brain lesion volume analysis was used for the neuronal cell-density determination.

### 2.12. Statistical Analysis

Experiments consisted of 10 mice in each group. Statistical analyses and data graphing were conducted utilizing computer software designed for scientific data analysis (GraphPad Prism 4; GraphPad Software Inc., La Jolla, CA, USA). Quantitative data were displayed as the mean with standard error of the mean and differences among means determined by one-way analysis of variance (*p* < 0.0001) and either a Bonferroni’s or Dunnett’s multiple comparison test for the data, respectively (*p* < 0.05).

## 3. Results

### 3.1. Food Intake and Weight Changes

Ten-week-old mice were fed diets supplemented with GrandFusion diets (2%) for 2 months prior to injury. The diets were as follows: Group 3 received a 2% GrandFusion (GF1, NF-216—Fruit and Veggie #1 Blend), with the ND; Group 4 received a 2% GrandFusion diet (GF2, NF-316—Fruit #2 Blend); and Group 5 received a 2% GrandFusion diet (GF3, NF-416—Vegetable #3 Blend). These are the same diets that were used in previous studies [14,15,16]. The animals were examined for food intake and body weight every week for the ten weeks of feeding. The mice on all diets maintained a constant intake of food over the course of the study. In addition, consistent with the food intake, all of the mice showed a similar gain in weight over the two months. 

### 3.2. Impact of Diet on Neuromotor Activity Following TBI

The mice were subjected to controlled cortical impact (CCI) as described previously [17]. Mice were examined for neuromotor function after the TBI using the rotarod test by measuring the length of time the animals were able to stay on a rotating rod before falling off (latency to fall); as the time decreased, the poorer the neuromotor score. Latency times to fall were measured just before trauma on day 0 (before injury) and on days one, three, and seven after injury (Figure 1). Before the injury, all groups had similar latency periods times (281.2 ± 3.7 sec). Sham mice maintained neuromotor function throughout the testing. On day 1 post-trauma, TBI alone mice had latency times of 84.5 ± 7.3, which was a 70% shorter time than the sham controls. TBI mice had poorer neuromotor function relative to controls. The TBI mice showed a slow recovery to about 136.7 ± 9.4 sec. Motor performance was evaluated for mice fed the GF diets (GF1, GF2, and GF3). In general, the mice on the GF diets showed significantly less neuromotor dysfunction and recovered faster than did TBI mice. On day 1, the latency times were 174.9 ± 16.1 (GF1), 153.5 ± 18.0 (GF2), and 164.7 ± 17.4 (GF3) sec, with TBI mice having a significant 45 to 52% shorter time than that of the treated animals. The latency times on days three and seven for the treated mice were significantly better than the TBI alone mice. Importantly, these data suggest that diets enriched in fruits and vegetables prior to injury significantly reduced the severity of neuromotor dysfunction from TBI, with near full recovery at 7 days after TBI.

### 3.3. Improved Cognitive Deficits with GF Diets

We determined whether GF diets were able to attenuate the long-term cognitive deficits as assessed by the Morris water maze as required to locate a platform submerged in water from 22 to 26 days’ post injury (Figure 2). As seen in the figure, there was a significant improvement in cognitive performance after CCI for all the groups of mice. Conversely, mice on the GF-enriched diets exhibited less cognitive dysfunction than those on a regular diet. The deficits in memory after CCI in mice on a regular diet were expressed as less time spent in the target quadrant. In addition, there were no differences in the swimming velocity suggesting no effect on enhanced or retarded physical parameters but reflect direct changes in cognitive functions.

### 3.4. Improvement in Sensorimotor Deficits Following TBI

To further assess the impact of the GF diets on sensorimotor deficits, we determined the outcomes via the wire hanging, cylinder, and grid walking tests. In the wire hanging test, following TBI, mice experienced a significant drop in scores compared to pre-TBI testing which was followed by a gradual recovery of function, whereas the sham mice exhibited no decrease. In comparison, mice fed with the GF diets showed better performance with significantly improved scores during the course of the first 14 days’ post-injury (Figure 3A). In addition, when the mice were subjected to the cylinder test, the mice on the diets had reduced limb dysfunction (Figure 3B). The mice on the GF diets showed an enhanced sensorimotor performance as demonstrated in the grid walking and foot-fault tests (Figure 3C,D). In summary, the GF diets attenuated the sensorimotor deficits seen following TBI.

### 3.5. Protection of the Brain with GF Diets Following TBI

Animals tested in the behavioral assays described above were sacrificed at the end of the study, and their brains were histopathologically evaluated for lesion volume with quantification (Figure 4). Brain lesions were absent in sham mice, but TBI mice had lesion volumes of 15.6 ± 3.2 cubic millimeters (mm^3^) (Figure 4B,F). Brain lesion volumes of mice treated with the GF diets were 4.9 ± 2.6 (GF1), 5.4 ± 3.0 (GF2), and 5.2 ± 2.4 (GF3) mm^3^ (Figure 4C,D,F). Notably, the GF diet-treated mice had about one third less lesion volume compared to the control animals, indicating that the diets provided protection from TBI lesion. Mouse brains were evaluated for neuronal cell densities in the CA3 region of the hippocampus by quantitative histopathology image analysis (Figure 5). The CA3 region of the hippocampus is vulnerable to TBI and has been shown to participate in cognitive impairment [25,26,27]. Neuronal cell densities from sham mice showed no decrease, while TBI mice had a significant loss of neuronal cell density (Figure 5). The effect on neuronal cell densities in the presence of the GF diets demonstrated an attenuation of the loss. Neuronal cell densities were significantly higher in the GF diets treated animals.

### 3.6. Diet-Induced Reduction in Neuroinflammation Following TBI

To determine the impact of the diets on neuroinflammation in the mouse brain after TBI, mouse brains were examined for the expression of inflammatory markers. We evaluated the levels of the cytokine’s tumor necrosis factor-α (TNF-α), interleukin-1β (IL-1β), and transforming growth factor-β (TGF-β) at 24 hours after TBI (Figure 6). As seen in Figure 6, the GF diets significantly reduced TNF-α, IL-1β, and TGF-β levels after injury. All the diets showed an effect reducing the above cytokine levels by 75% (TNF-α), 89% (IL-1β), and 87% (TGF-β).

### 3.7. Altered Cathepsin B Activity in Diet Treated TBI Mice

Our previous studies have shown that TBI results in an increase in cathepsin B protein and activity that can lead to inflammatory mediators such as IL-1β [17]. To determine the mechanisms associated with the increase in inflammation (Figure 6), we determined the impact of the GF diets on cathepsin B protein and activity. TBI increased cathepsin B levels in the brain and the GF diets reduced or attenuated the increase (Figure 7). These results suggest that reduction in inflammation occurring with treatments was partially the result of inhibition of cathepsin B activity. The brains of mice evaluated for cathepsin B activity and protein levels (Figure 7) were also evaluated for pro-apoptotic Bax protein 24 h after trauma (Figure 8). Western blot analysis demonstrated that sham mice had low levels of Bax protein, whereas TBI mice had Bax expression levels that were a significant 15-fold greater. GF treated mice resulted in a significant decrease in Bax levels, showing that the diets reduced Bax levels by a significant ~85%, relative to TBI mice.

## 4. Discussion

In the present study, we examined the impact of diets rich in vegetables and/or fruits on outcomes and recovery/repair from traumatic brain injury (TBI). We found that long-term feeding of these diets for 2 months prior to injury improved behavioral outcomes, reduced inflammation, and diminished lesion volume in a mouse model of TBI. 

Severe traumatic brain injury (TBI) is one of the most common causes of death in young adults in the industrialized nations [28,29,30]. Secondary injury which is the result of the initiation of pathophysiological signaling pathways radically exacerbates the principal injury triggered by the trauma and is accountable for almost a third of all deaths associated with traumatic brain injury [31,32,33]. Initial treatment for patients with TBI targets the secondary injury to minimize the injury and additional loss of “healthy” brain tissue, to limit the stimulation of pro-inflammatory mechanisms, and to minimize the consequences of these cascades [34,35,36]. The limited approach to therapeutic strategies includes continuous monitoring of intracranial pressure, arterial hypotension, hypoxemia, and thromboembolic complications following TBI [37,38]. The impact of nutrition and the effects on the short- and long-term outcomes of mild and severe traumatic brain injuries have been disregarded for a long time [39,40]. There is moderate evidence to date, that nutritional supplementation should be initiated within the first 24 h following a TBI [41,42]. A number of studies have shown that oral consumption of nutritional support or supplements may have significant benefit to the individual [43]. Several nutrients that have shown preliminary promise as aids in treating traumatic brain injury, in particular, choline, creatine, *n*-3 fatty acids, and zinc [44,45,46,47,48]. In addition, if full nutritional support (caloric intake) is initiated immediately following the injury, there is a significant reduction in the impact of infections and overall complications [49,50].

The establishment of adequate nutritional support for patients with TBI has been a critical issue for many years [51]. The primary and secondary injuries in TBI, initiate metabolic imbalances that impact methods of administration, timing, and dosing of nutritional support, as well as therapeutics [52]. Nutritional support should not only include the application of fluids, electrolytes, drugs, glucose, and other feeding regiments but needs to allow for monitoring of these entities to prevent treacherous fluctuations that could endanger the patient [53]. Studies have shown that TBI patients can endure feeding via the small bowel, however, in the long-term patients need to convert to gastric feeding to allow for rehabilitation [54,55]. Individuals need to be monitored for dysphagia and other negative outcomes that can affect the quality of life and appropriate oral nutrition [56,57]. The influence of early nutrition support may be the critical factor for promising outcomes following TBI.

Our previous studies have shown that nutritional supplementation provided by the GF diets helped to improve outcomes from neurological diseases, attenuate changes in age-related deficiencies and improve physical consequences [14,15,16]. We demonstrated that feeding of the GF diets to mice before cerebral ischemia, limited the injury and improved behavioral outcomes [14]. Our results, and others, indicate that the diets were able to minimize inflammatory mediators and oxidative damage as well as enhance neuronal recovery [58,59,60,61,62,63,64,65,66,67]. Dietary supplementation in an aged rat model demonstrated a reversal of age-related phenomenon of elevated inflammation and reactive oxygen species (ROS) and enhanced the physical activity in older animals [15]. Finally, we recently showed that the GF diets boosted physical stamina in young animals [16]. These data suggest that supplementation with vegetables and fruits and the associated components are necessary and sufficient to provide protection from injury and to stimulate performance in animal models.

## 5. Conclusions

In conclusion, this study established that prophylactic long-term treatment of mice to diets enriched with vegetable and/or fruit concentrates attenuated TBI when administered prior to the injury. This demonstrated that diets containing anti-oxidants, anti-inflammatory agents, and other compounds that are efficacious in a mouse model of TBI may provide a potential preventative treatment. Therefore, since the brain is modifiable, adaptable, and regenerative in nature, approaches that retard inflammation and oxidative stress, and stimulate regenerative processes are viable approaches to temper damage from brain injury.

## Figures and Tables

**Figure 1 nutrients-11-00299-f001:**
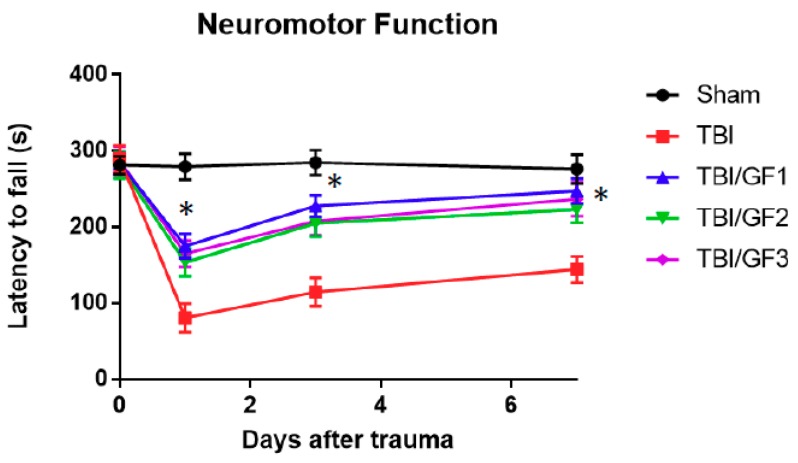
Effects of GrandFusion^®^ (GF) diets on neuromotor dysfunction. Neuromotor dysfunction was assessed at different time points during the week after traumatic brain injury (TBI) using the rotarod assay by measuring latency to fall time, with a shorter time reflecting a greater dysfunction. Latency to fall times for the sham, TBI, and TBI plus GF diet mice are shown. Mice were fed a normal diet or diets supplemented with 2% GF. Each point represents mean +/− SD (*n* = 10 per time point). * *p* < 0.001 compared to TBI group.

**Figure 2 nutrients-11-00299-f002:**
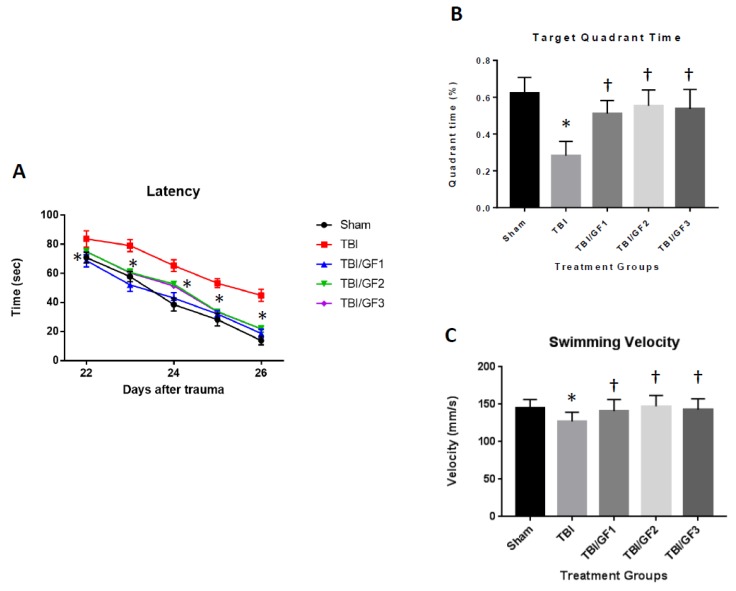
Effects of GF diets on Morris water maze testing. (**A**) Latency, (**B**) time in target quadrant, and (**C**) swimming velocity in sham, TBI and TBI + diet mice. Mice were subjected to the Morris water maze and examined for the time to find the platform and the time spent in the target quadrant. No difference was detected in the swimming velocity. Data are expressed as the mean +/− SD (*n* = 10, A = * *p* < 0.001 compared to TBI group; B/C = * *p* < 0.001 compared to sham group, † *p* < 0.001 compared to TBI group).

**Figure 3 nutrients-11-00299-f003:**
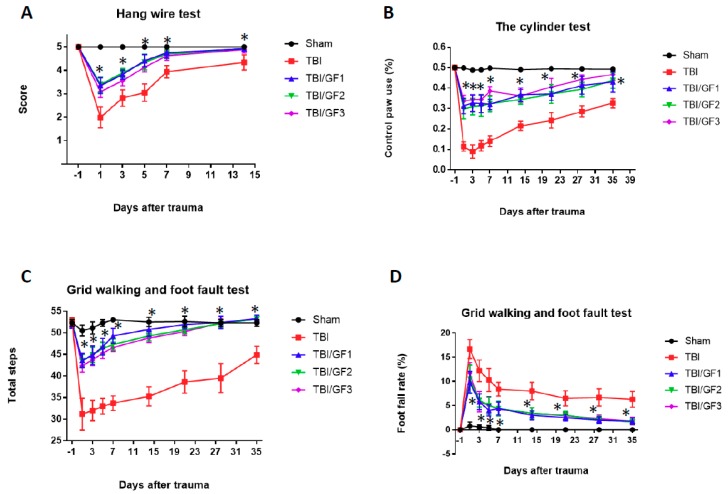
Effect of GF diets on short- and long-term behavioral deficits caused by TBI. Performance comparison for the first 14 days’ post-injury between mice on the different diets; (**A**) hanging wire test demonstrating the ability of mice to hang on to a wire after TBI; (**B**) cylinder test showing the effect of diets on the ability to function with their contralateral paw; (**C**,**D**) grid walking and foot fault test to determine the impact of diets on foot fault rate and step frequency. The results are expressed as the mean +/- SD (*n* = 10, * *p* < 0.001 compared to the TBI group).

**Figure 4 nutrients-11-00299-f004:**
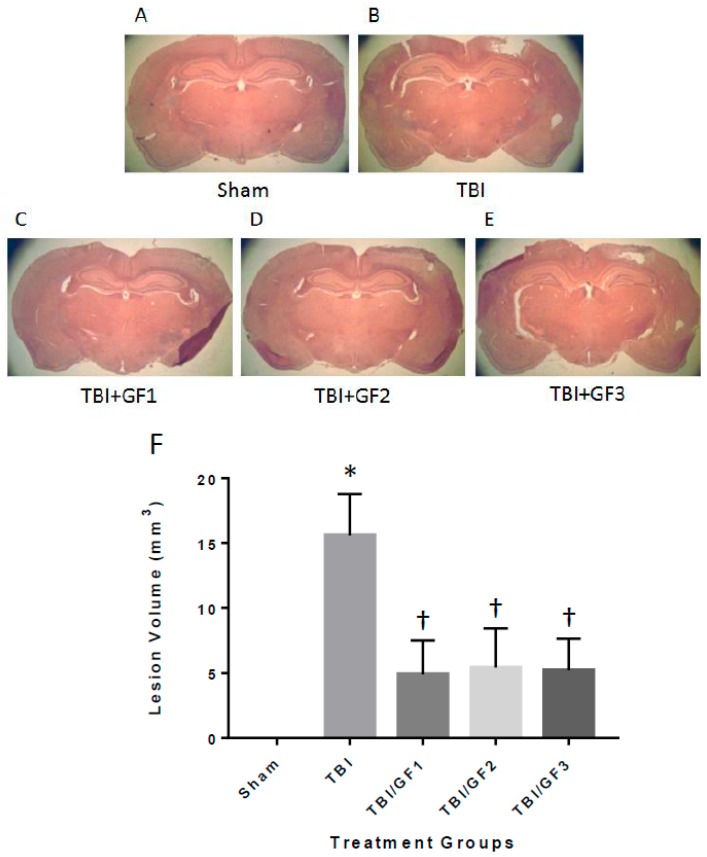
GF diet suppresses lesion volume in TBI induce mice. (**A**–**E**) Mice fed control or GF diets were euthanized at the end of the experiment, and their brains were analyzed by quantitative histology to determine the brain lesion volume at the impact site. (A) Sham; (B) TBI; (C) TBI + GF1; (D) TBI + GF2; and (E) TBI + GF3. (**F**) Brains were analyzed by quantitative histology to determine the brain lesion volume at the impact site. The results are expressed as mean +/- SD (*n* = 10, * *p* < 0.001 compared to the sham group; † *p* < 0.001 compared to the TBI group).

**Figure 5 nutrients-11-00299-f005:**
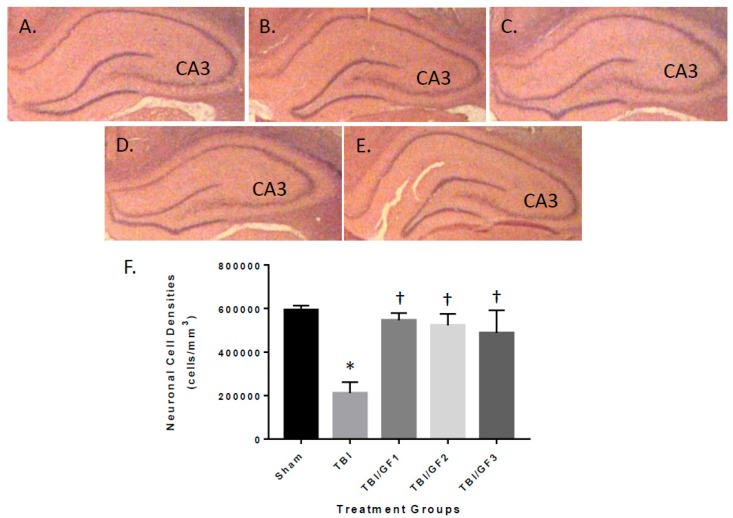
TBI-mediated reduction in hippocampal neuronal densities. (**A**–**E**) Brains of mice were analyzed in previous studies were evaluated at the end of the study for neuronal densities in the CA3 region of the hippocampus. (A) Sham; (B) TBI; (C) TBI + GF1; (D) TBI + GF2; and (E) TBI + GF3. (**F**) Regions were analyzed by quantitative histology to determine the brain lesion volume at the impact site. The number of neuronal cells/mm^3^ was determined in each of the mouse groups. The results are expressed as mean +/- SD (*n* = 10, * *p* < 0.001 compared to the sham group; † *p* < 0.001 compared to the TBI group).

**Figure 6 nutrients-11-00299-f006:**
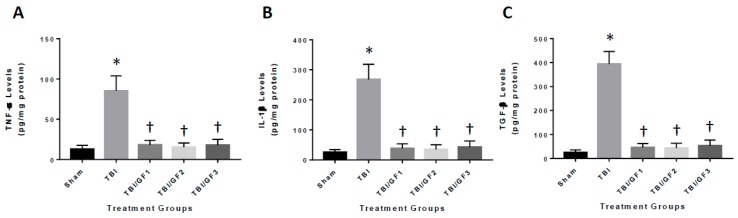
Reduced inflammatory markers in the brain after TBI. Mice were sham, TBI, or TBI subjected to various diets followed by 24 hours of recovery. Quantitative analysis of IL-1β (**A**), TNF-α (**B**), and TGF-β (**C**) in the TBI brain was determined by ELISA. Brain homogenates were subjected to ELISA. The results are expressed as mean +/- SD (*n* = 10, * *p* < 0.001 compared to the sham group; † *p* < 0.001 compared to the TBI group).

**Figure 7 nutrients-11-00299-f007:**
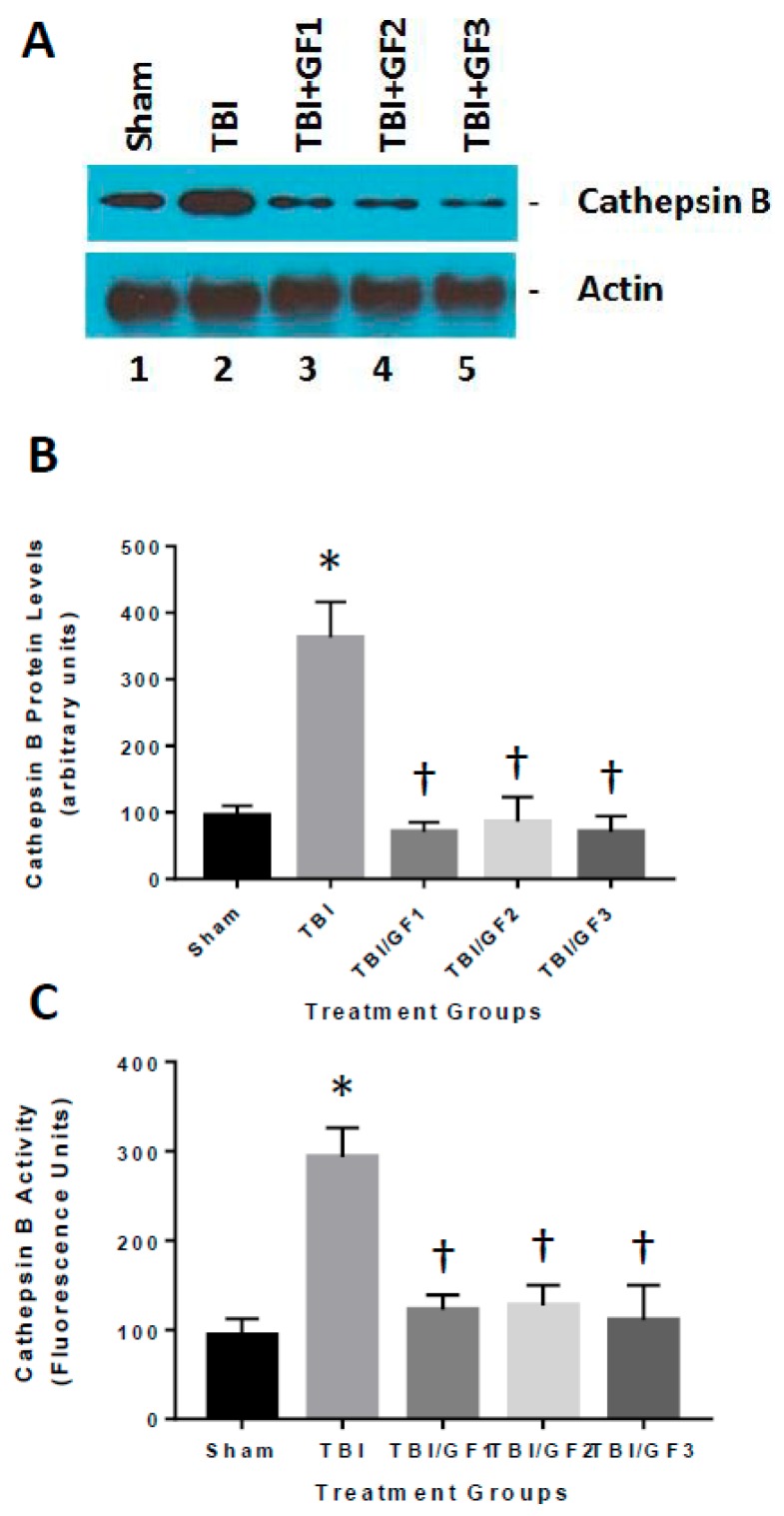
The effect of GF diets on cathepsin B activity. (**A**) Brain cathepsin B protein levels were determined 24 hours following TBI. Western blot analysis of the cathepsin B levels in the brains of sham, TBI, and TBI + GF diets. (**B**) Quantitative analysis of cathepsin B protein levels of the mice in A. (**C**) Brain cathepsin B activities were determined in the mice following 2 hours after TBI in sham, TBI, and TBI + GF diets. The results are expressed as mean +/- SD (*n* = 10, * *p* < 0.001 compared to the control group; † *p* < 0.001 compared to the TBI group).

**Figure 8 nutrients-11-00299-f008:**
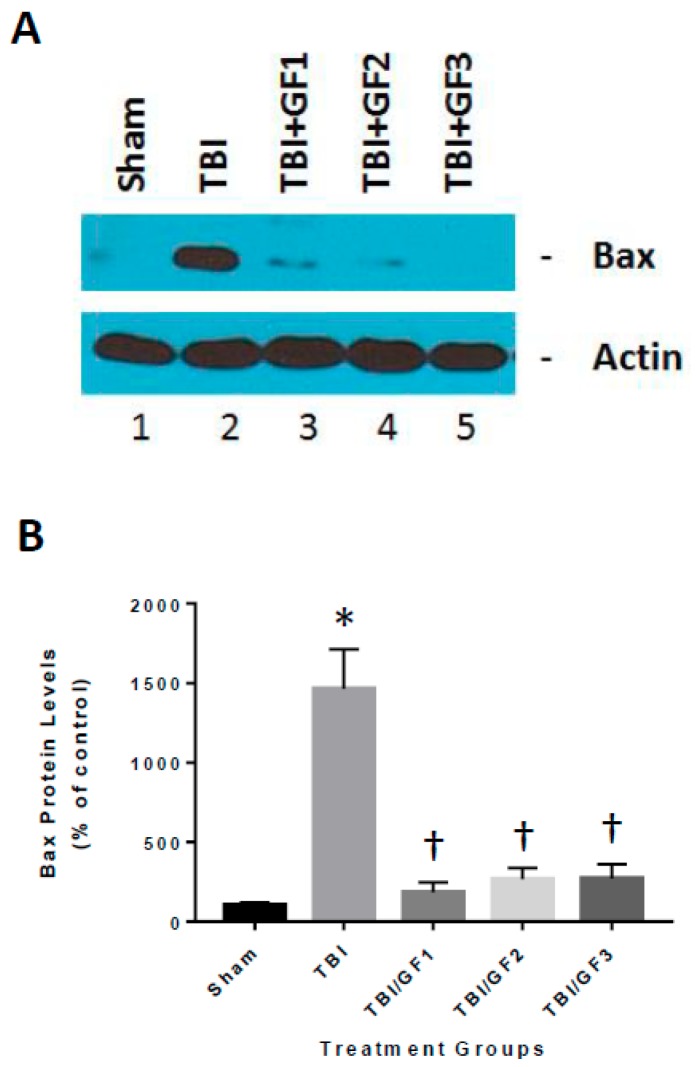
TBI-induced increase in BAX levels in brains of mice with and without GF diets. The brains of the animals subjected to TBI and diets were examined for proapoptotic BAX protein levels by Western blot analysis (**A**) and quantitative densitometry (**B**). The results are expressed as mean +/- SD (*n* = 10, * *p* < 0.001 compared to the sham group; † *p* < 0.001 compared to the TBI group).

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
