# Peer review of "Reduced Neuroinflammation and Improved Functional Recovery after Traumatic Brain Injury by Prophylactic Diet Supplementation in Mice"

_nutrients, 2019, doi:10.3390/nu11020299_

Reviewer 1 Report

In this manuscript, the authors investigated the effects of pre-TBI treatment of GrandFusion (GF) diet on post-injury cognition, motor function, inflammation, lesion volume, and proapoptotic markers in a mouse model of TBI. Their results showed that pre-injury GF diet treatment improved cognitive and motor function, limited the extent of the injury and reduced inflammation after CCI. They concluded that long term pre-injury GF diet treatment can reduce TBI and may provide a preventive treatment for TBI.

Here are my comments.

 If the authors aim to explore new approaches for treatment of TBI which occurrence is unpredictable, it would be more meaningful to test the effects of post-injury GF diet treatment rather than pre-injury treatment?

The experimental animals have received 2-month GF diet treatment prior to CCI. What are the effects of the treatment on those animals before the injury occurred? Could the effects observed after the injury were already there before the injury?

It seems that the GF diet treatment started when mice were 2-4 weeks old even before they were weaned. Why were such young mice used when their brains were still under critical development? How could those young mice be gavaged with GF diet when they were supposed to be breastfed?

Regarding CCI procedure, dwelling time of the probe on the brain is another important parameter that can affect severity of the injury. It should be included in the methods.

Font of some texts in figures is too small and need to be adjusted.

Author Response

Thank you for the review.  We have addressed the comments below.

1) The goal of the project was to determine the impact of diet on TBI outcomes not as a treatment.  Subsequent studies will evaluate that effect.  We agree that the TBI is unpredictable, so a diet enriched in vegetables and fruits can help limit the extent of the injuries.

2) Since the animals do not show symptoms of TBI prior to the injury, the effects of nutrients would be limited.  However, we have published the effects in aging.

3) The animals were actually 10-12 weeks of age when the study was started.  

4) Added.

5) Changed.

Reviewer 2 Report

The paper is dedicated to determining the influence of diets rich in vegetables and fruits on the outcomes associated with traumatic brain injury (TBI). Mice were fed diets enriched in fruits and vegetables for 2 months and then subjected to TBI. Authors demonstrated that long-term treatment of mice to diets enriched with vegetable and/or fruit concentrates attenuated TBI when administered prior to the injury. Authors looked for neuromotor function, lesion volume, neuronal density in the hippocampus and reduced inflammation.

The paper has a scientific rationality but it needs to be improved.

First of all I do not read Table 1 that has been cited by authors (page 2, line 78). Again, I do not read the number of rats composing each group.

The sections 2.10 and 2.11 (Brain lesion volume analysis and Neuronal cell density determination) are too hasty and do not allow to understand the subsequent results. In fact, section 3.5 (Protection of the brain with GF diets following TBI) does not allow to understand the morphological and histological results that must be reported together with more detailed images of both brain volume and neuronal density. In this regard, the authors in section 2.10 write that "Every fourth section was processed for immunohistochemical analysis beginning from a random start point", but I do not read either the methods or the results in detail. Again, microscopic images of immunohistochemical results must be presented by the authors.

Figures 1, 2A, 3, are incomprehensible in size and color. These images do not allow to discriminate the results.

The word alpha and the word beta (TNF ?, IL1?) are constantly absent in the text.

In the references the authors are very self-cited but many recent articles dedicated to TBI and oxidative stress (reviews) are missing.

1.Diffuse Axonal Injury and Oxidative Stress: A Comprehensive Review. Frati A, Cerretani D, Fiaschi AI, Frati P, Gatto V, La Russa R, Pesce A, Pinchi E, Santurro A, Fraschetti F, Fineschi V. Int J Mol Sci. 2017 Dec 2;18(12). pii: E2600. doi: 10.3390/ijms18122600. Review.

2.Oxidative stress in traumatic brain injury.Rodríguez-Rodríguez A, Egea-Guerrero JJ, Murillo-Cabezas F, Carrillo-Vico A. Curr Med Chem. 2014 Apr;21(10):1201-11. Review.

3.Connexin40 correlates with oxidative stress in brains of traumatic brain injury rats. Chen W, Guo Y, Yang W, Zheng P, Zeng J, Tong W. Restor Neurol Neurosci. 2017;35(2):217-224. doi: 10.3233/RNN-160705.

4.Huperzine A alleviates neuroinflammation, oxidative stress and improves cognitive function after repetitive traumatic brain injury. Mei Z, Zheng P, Tan X, Wang Y, Situ B. Metab Brain Dis. 2017 Dec;32(6):1861-1869. doi: 10.1007/s11011-017-0075-4. Epub 2017 Jul 26.

5.Lack of mitochondrial ferritin aggravated neurological deficits via enhancing oxidative stress in a traumatic brain injury murine model. Wang L, Wang L, Dai Z, Wu P, Shi H, Zhao S. Biosci Rep. 2017 Nov 6;37(6). pii: BSR20170942. doi: 10.1042/BSR20170942. Print 2017 Dec 22.

6.Elevation of oxidative stress indicators in a pilot study of plasma following traumatic brain injury. Halstrom A, MacDonald E, Neil C, Arendts G, Fatovich D, Fitzgerald M. J Clin Neurosci. 2017 Jan;35:104-108. doi:10.1016/j.jocn.2016.09.006. Epub 2016 Sep 30.

7.Interactions of oxidative stress and neurovascular inflammation in the pathogenesis of traumatic brain injury. Abdul-Muneer PM, Chandra N, Haorah J. Mol Neurobiol. 2015;51(3):966-79. doi: 10.1007/s12035-014-8752-3. Epub 2014 May 28. 8.Salubrinal reduces oxidative stress, neuroinflammation and impulsive-like behavior in a rodent model of traumatic brain injury. Logsdon AF, Lucke-Wold BP, Nguyen L, Matsumoto RR, Turner RC, Rosen CL, Huber JD. Brain Res. 2016 Jul 15;1643:140-51. doi: 10.1016/j.brainres.2016.04.063. Epub 2016 Apr 27.

9.NADPH Oxidase 2 Regulates NLRP3 Inflammasome Activation in the Brain after Traumatic Brain Injury. Ma MW, Wang J, Dhandapani KM, Brann DW. Oxid Med Cell Longev. 2017;2017:6057609. doi: 10.1155/2017/6057609. Epub 2017 Jul 12.

10.Administration of Protocatechuic Acid Reduces Traumatic Brain Injury-Induced Neuronal Death. Lee SH, Choi BY, Lee SH, Kho AR, Jeong JH, Hong DK, Suh SW. Int J Mol Sci. 2017 Nov 23;18(12). pii: E2510. doi: 10.3390/ijms18122510.

Author Response

See below.  We have spell checked the manuscript.

1) Table 1 is now included.  Mice were used in the study and the total number is now included.

2) We have clarified the information presented.  We have added the figures for the neuronal cell density.

3) We have helped to clarify the figures.

4) Corrected the alpha in the text.

5) added the references indicated by the reviewer.

Round  2

Reviewer 1 Report

The manuscript has improved after revision.

However, please edit the manuscript further, including its title, abstract, and main text,  to make it clear that the treatment was given before injury and the effects resulted from the pre-injury treatment.

Author Response

We have now added in title, abstract and text that the diets were used either prior to injury or as a prophylaxis.  

Reviewer 2 Report

Authors did not perform all the requested changes.

First of all I do not read Table 1 that has been cited by authors (page 2, line 79).

The sections 2.10 and 2.11 (Brain lesion volume analysis and Neuronal cell density determination) are too hasty and do not allow to understand the subsequent results. In fact, section 3.5 (Protection of the brain with GF diets following TBI) does not allow to understand the morphological and histological results that must be reported together with more detailed images of both brain volume and neuronal density. 

Figures 1, 2A, 3, are incomprehensible in size and color. These images do not allow to discriminate the results.

The word alpha and the word beta (TNF ?, IL1?) are constantly absent in the text and figure legend (see page 9.

Author Response

The Table 1 is provided in supplementary materials. 

We have included the brain regions with arrows to identify the regions of interest.  We have included text to better describe the neuronal loss.

We have now added color to the figures.

We have corrected everywhere to ensure the correct symbols for the cytokines.  If they still are present, they will be corrected in the proofs.